# Transcriptome of the parasitic flatworm *Schistosoma mansoni* during intra-mammalian development

Arporn Wangwiwatsin[1,2], Anna V. Protasio[1,3], Shona Wilson[3], Christian Owusu[1], Nancy E. Holroyd[1], Mandy J. Sanders[1], Jacqueline Keane[1], Mike J. Doenhoff[4], Gabriel Rinaldi[1], Matthew Berriman[1]*

1 Wellcome Sanger Institute, Wellcome Genome Campus, Hinxton, United Kingdom, 2 Department of Biology, Faculty of Science, Khon Kaen University, Khon Kaen, Thailand, 3 Department of Pathology, Tennis Court Road, University of Cambridge, Cambridge, United Kingdom, 4 School of Life Sciences, University of Nottingham, University Park, Nottingham, United Kingdom

* mb4@sanger.ac.uk

**Data Availability Statement:** All sequencing data are available from the European Nucleotide Archive

## Abstract

Schistosomes are parasitic blood flukes that survive for many years within the mammalian host vasculature. How the parasites establish a chronic infection in the hostile bloodstream environment, whilst evading the host immune response is poorly understood. The parasite develops morphologically and grows as it migrates to its preferred vascular niche, avoiding or repairing damage from the host immune system. In this study, we investigated temporal changes in gene expression during the intra-mammalian development of *Schistosoma mansoni*. RNA-seq data were analysed from parasites developing in the lung through to egg-laying mature adult worms, providing a comprehensive picture of *in vivo* intra-mammalian development. Remarkably, genes involved in signalling pathways, developmental control, and adaptation to oxidative stress were up-regulated in the lung stage. The data also suggested a potential role in immune evasion for a previously uncharacterised gene. This study not only provides a large and comprehensive data resource for the research community, but also reveals new directions for further characterising host–parasite interactions that could ultimately lead to new control strategies for this neglected tropical disease pathogen.

## Author summary

The life cycle of the parasitic flatworm *Schistosoma mansoni* is split between snail and mammalian (often human) hosts. An infection can last for more than 10 years, during which time the parasite physically interacts with its mammalian host as it moves through the bloodstream, travelling through the lungs and liver, to eventually establish a chronic infection in the blood vessels around the host gut. Throughout this complex journey, the parasite develops from a relatively simple larval form into a more complex, sexually reproducing adult. To understand the molecular basis of parasite interactions with the host during this complex journey we have produced genome-wide expression data from developing parasites. The parasites were collected from experimentally-infected mice over

(accession number ERP113121). Processed count data are available within the manuscript and its Supporting Information files.

**Funding:** AW, AVP, CO, NEH, MS, JK, GR, MB were funded by Wellcome (wellcome.ac.uk grant 206194). AW was also funded by the Government of Thailand, Ministry of Science and Technology (mhesi.go.th/main/en). SW and MD received no specific funding for this work. The funders had no role in study design, data collection and analysis, decision to publish, or preparation of the manuscript.

**Competing interests:** The authors have declared that no competing interests exist.

its developmental time-course from the poorly studied lung stage, to the fully mature egg-laying adult worm. The data highlight many genes involved in processes known to be associated with key stages of the infection. In addition, the gene expression data provide a unique view of interactions between the parasite and the immune system in the lung, including novel players in host-parasite interactions. A detailed understanding of these processes may provide new opportunities to design intervention strategies, particularly those focused on the early stages of the infection that are not targeted by current chemotherapy.

## Introduction

The blood fluke *Schistosoma mansoni* is a major aetiological agent of hepatic and intestinal schistosomiasis, a Neglected Tropical Disease that affects over 200 million people around the world, largely in developing regions [1]. Standard treatment of schistosomiasis relies on a single drug, praziquantel, and drug resistance is an ever-present threat with emerging reports of reduced efficacy to praziquantel in the field [2]. A better understanding of the molecular and cellular mechanisms underlying the establishment of infections could reveal potential vulnerabilities for targeting in new control tools.

Susceptible aquatic snails transmit schistosomes between mammalian hosts by releasing infectious larvae called cercariae that seek out a mammalian host, penetrate through its skin and transform into a juvenile form known as schistosomula. The schistosomula enter the bloodstream and by day 6 post-infection are mainly present in the lung capillaries [3,4]. The parasites subsequently circulate within blood vessels throughout the body while developing organs, growing in size and eventually reaching the liver. In the case of *S. mansoni*, male and female parasites pair up after approximately 28 days post-infection and migrate from the liver to the mesenteric veins [3]. By day-35 the parasites reach full maturity in the mesenteric veins [3], where they release eggs that traverse the intestinal wall, reach the gut lumen, and pass with faeces into the environment.

Large-scale transcriptional changes have been extensively described across the life-cycle of *Schistosoma* spp. but in many cases ESTs, SAGE or microarray approaches were used [5–12] that lack the dynamic range and sensitivity of more contemporary RNA-seq approaches. Where RNA-seq has been used, the focus has been on just a few developmental stages. Examples include a focus on eggs, the drivers of schistosome-induced pathology, and the transformation of the free-living cercariae into schistosomula, where extraordinary morphological and physiological changes mark the start of intra-mammalian development [13,14]. These and other studies have produced new lists of abundantly transcribed factors including numerous G-protein coupled receptors (GPCRs) [15], as well as confirming previous biochemical observations, such as the switch to anaerobic metabolism as the parasite's physiological environment changes [16,17]. Studies in both *S. mansoni* and *S. japonicum* have shown high expression of genes involved in nutrient acquisition and oxidative stress responses in adult worms [8,9,11]. Prominent components and processes of reproductive development and egg production have also been discovered but in many cases the primary motivation has been to identify tentative drug and vaccine targets, such as tetraspanins and GPCRs [10]. Specific comparisons of males and females from single-sex and mixed-sex infections have provided more insight into reproductive development and male-female interactions, including the prominent roles of neuropeptide signalling and stem cell differentiation [18].

In contrast to adult stages, gene expression in early stages of intra-mammalian development has been poorly studied. Studying several specific developmental stages has been compromised

by difficulties in obtaining parasite material. This has resulted in a complex picture being pieced together using material obtained *in vivo* and *in vitro* [10,17]. Given the enormous differences between the *in vivo* and *in vitro* environment, gene expression is likely to be dramatically affected, particularly amongst those genes related to nutrient acquisition, stress responses and immune evasion [8]. The lung stage, in particular, is a key point of attrition during primary and challenge infections of animal models [4,19,20].

In order to fill these knowledge gaps, we have analysed the *S. mansoni* transcriptome during six discrete stages of development within experimentally-infected mice, ranging from the early lung stage to sexually mature egg-laying worms. Gene expression changes correlated well with the well-described biology of egg production in the adult stages, and with growth and developmental control in the liver developmental stages of the parasite. Notably, host immune-evasion and oxidative stress regulation by the parasite, were clearly evident in the lung-stage transcriptome in addition to potential novel players in host-parasite interactions. This study not only provides a rich and systematic resource for the community to understand schistosome biology, but also reveals new molecular mediators at the host-parasite interface.

## Materials and methods

### Ethical statement

All procedures involving mice were performed by authorised personnel according to the UK Animals (Scientific Procedures) Act 1986 with project license held by MJD (number PPL 40/3595). The work was approved by the Ethical Review Committee of the University of Nottingham and was carried out in strict accordance with UK Home Office regulations for animal welfare and amelioration of suffering.

### Mouse infection

Mice were infected with cercariae of *S. mansoni* (Puerto Rican strains) shed from *Biomphalaria glabrata* snails as described [21]. In brief, percutaneous infections were performed by applying suspensions of mixed-sex cercariae to the shaved abdomens of anaesthetised mice and leaving for 30 minutes. Mice were infected with the following numbers of cercariae: 2000 to provide day-6 and day-13 parasites, 500 for days 17 and 21, 350 for day 28, and 300 for day 35 (Fig 1). Four mice were used for adult stage parasites and up to eight mice were used for juvenile stages (Fig 1). More mice were used for early time points due to the greater uncertainty in acquiring the samples. All mice were females of CD-1 outbred strain (Charles River, Harlow, UK) aged between 8–12 weeks by the time of infection. A pool of mixed-sex parasites from each mouse was considered a biological replicate.

### Parasite material collection and imaging

On the indicated day post-infection, mice were culled using an overdose of pentobarbitone containing 10 U/ml heparin. Day-6 lung-stage parasites were collected as described [21]. Briefly, lungs collected from infected mice were minced using a sterile scalpel cut into approximately 1 mm$^3$ pieces and incubated in 50 ml heparinised modified Basch media (10 U/ml heparin, 1 mg/ml Lactalbumin hydrolysate powder, 0.5 µM Hypoxanthine, 0.008 mg/ml Insulin, 1 µM Hydrocortisone, 0.2 µM Triiodothyronine, 0.5X MEM Vitamins, 5% Schneiders Drosophila Medium, 1% Hepes Buffer, 10% Fetal bovine serum, 2% Antibiotic-Antimycotic in DMEM) for approximately 1 hour at room temperature, followed by 3 hours at 37˚C, 5% $CO_2$ to allow the parasites to exit the tissue. The tube contents were mixed by turning the tube 2–3 times before being passed through a 600 µm mesh into new 50 ml tubes to separate large pieces

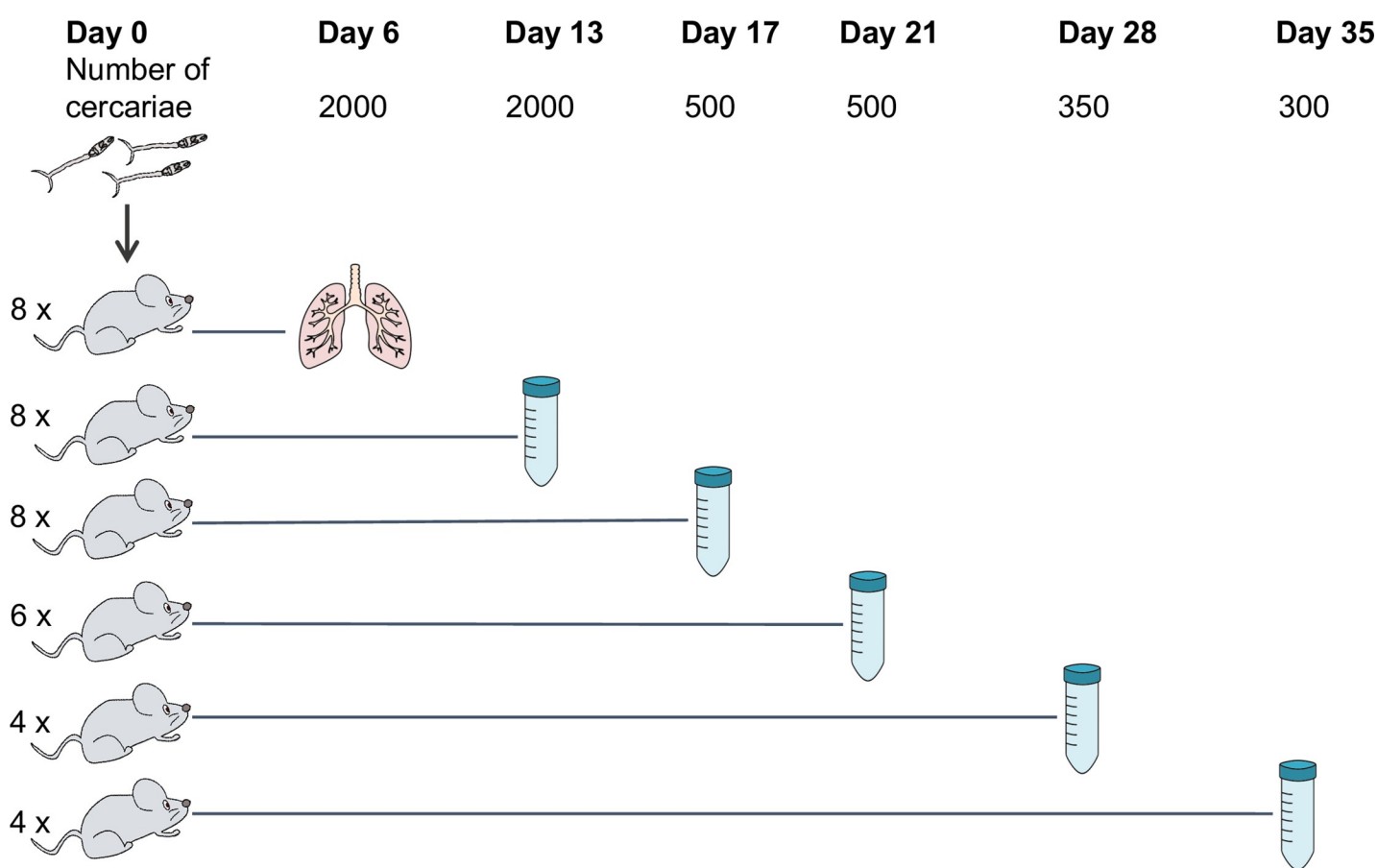

**Fig 1. The numbers of mice and cercariae used for infections.** Mice were infected with indicated numbers of cercariae for parasite collection at six time points post-infection. The number of mice used for each time point is shown on the left. The method for parasite collection at day 6 involves mincing and incubation of the lung. Collections at other time points were done by portal perfusion.

of tissue from worms. The filtrate was centrifuged at 150 x g for 3 minutes at room temperature and approximately half of the supernatant was discarded by gently decanting. Lung-stage worms were recovered using a Pasteur pipette to collect approximately 1–1.5 ml from the bottom of the tube. Given that parasites from day-6 post-infection were collected only from the lung and not by perfusion, circulating parasites that had left the lung were therefore excluded. For this reason, day-6 worms in the present study are hereafter referred to as 'lung stage'.

For all other time points, parasites were collected by portal perfusion with approximately 30 ml of perfusion media (Dulbecco's Modified Eagle's Medium (DMEM) high glucose, with 10 U/ml heparin). Parasites were left to settle for 30 minutes at room temperature and washed twice with DMEM before being recovered with a Pasteur pipette from the bottom of the tube.

Parasites were transferred to a Petri dish for imaging using an Olympus SZ61dissecting microscope with a Euromex Cmex10 camera and Image Focus 4.0 software. A subset of the parasites from each mouse was imaged for morphology scoring. After imaging, parasites were transferred into a 2 ml Eppendorf tube and centrifuged at 150 x g for 3 minutes before replacing the supernatant with 1 ml TRIzol (Thermo Fisher), left at room temperature for up to 1 hour, transferred to dry ice for transport, and later stored at -80 ºC until RNA extraction. An average of 31 worms were imaged per mouse (range 18–75 worms); this did not represent a total number of parasites collected.

## Parasite morphology scoring

Parasite morphology was classified using numerical scores based on published categories [22]. Parasite features used for categorisation were the presence of a haemozoin-filled gut, the shape and length of the gut (fork end, closed end, proportion of the posterior end of the gut to the anterior section), and whether the worms were paired or unpaired. The scoring was performed blindly, i.e. all images were renamed to randomised numbers. Different numbers of worms ranging from 18–75 were morphologically scored among replicates; hence, the number of parasites that fell into each morphology category was shown as a percentage of the total worms morphologically scored in that replicate.

## RNA extraction

RNA was extracted from parasite material using a modified phenol-chloroform method and column purification. Briefly, frozen samples in TRIzol reagent were thawed on ice, resuspended by gently pipetting and transferred to 2 ml tubes containing ceramic beads (MagNA Lyser Green Beads, Roche). The parasites were homogenised in a MagNA Lyser Instrument (FastPrep-24) at maximum speed twice for 20 seconds, with a 1-minute rest on ice in between. Next, 200 µl of chloroform-isoamyl alcohol 24:1 was added to each tube, followed by vigorous shaking for 5 seconds. The tubes were centrifuged at 13,000 x g for 15 minutes at 4˚C to separate the aqueous and organic solvent layers. The aqueous layer was transferred into a RNase-free 1.5 ml tube and one volume of 100% ethanol added and mixed by pipetting. The mixture was transferred to Zymo RNA Clean & Concentrator-5 column (Zymo Research) and processed according to the manufacturer's protocol. To elute the RNA, 15 µl of RNase-free water was added to the column and centrifuged for 30 seconds at 13,000 x g. The RNA concentration and integrity were measured by Agilent RNA 6000 Nano kit (Agilent Technologies), and its purity assessed using a NanoDrop spectrophotometer.

## Library preparation and sequencing

One to 2.8 µg of RNA was used to prepare each sequencing library. The libraries were produced using TruSeq Stranded RNA Sample Preparation v2 Kits (Illumina). Libraries were amplified using 10–14 cycles of PCR and cleaned using Agencourt AMPure XP Beads (Beckman Coulter). The libraries were quantified by qPCR before sequencing using the Illumina HiSeq 2500 platform. All sequencing data were produced as 75 bp paired-end reads and are available through ENA study accession number ERP113121.

## Read mapping and quantifying read counts

Sequencing reads (S1 File) were mapped to the *S. mansoni* reference genome v5 [17] from WormBase ParaSite [23] using TopHat version 2.0.8 [24] with default parameters except the following: -g 1 (only report 1 alignment for each read); -- library-type fr-firststrand (for dUTP Illumina library); -a 6 (minimum anchor length); -i 10 and -- min-segment-intron 10 (minimum intron length); -I 40000 and -- max-segment-intron 40000 (maximum intron length); -- microexon-search (find alignment to micro-exons). The resulting BAM files of accepted hits were sorted by read name (-n option) and indexed using SAMtools [25]. A GFF file of gene annotations from GeneDB.org was filtered to keep only the longest transcript for each gene (S2 File). The GFF file and sorted BAM files were used as inputs for HTSeq-count version 0.7.1 [26] to obtain read counts per transcript and used for read count analysis (S3 File). HTSeq-count was run with default parameters except with strand option set to suit dUTP libraries (-s reverse), and alignment score cut-off increased (-a 30).

## Differential expression analysis

Analyses were performed using RStudio version 0.99.489 [27], with R version 3.3.1 [28]. DESeq2 (version 1.12.3) [29] was used to import read counts, to investigate overall transcriptomic differences among samples using principal component analysis (PCA), to normalise read counts, and to identify genes differentially expressed in the time-course or in pairwise comparisons. PCA used regularized log–transformed (rlog-transformed) read count data as input. Differential expression analyses were performed with either likelihood-ratio tests (when the whole time-course was considered) or with the Wald test (when used with pairwise comparisons) and statistical significance was determined by adjusting p-values according to the Benjamini–Hochberg procedure to control false discovery rate [30]. Differentially expressed genes were defined as those with adjusted p-value $< 0.01$ and $\log_2$ fold change in expression $> 1$ or $< -1$.

## Gene clustering

Genes were clustered using self-organising maps constructed in the R package Kohonen (version 2.0.19) [31] based on their mean-normalised, rlog-transformed counts over the time-course. The mean-normalised counts were used to calculate means of replicates at each time point for each gene and used as input for clustering. Genes were grouped based on their expression pattern into 96 clusters. To reduce background signal, only genes that were differentially expressed in at least one time point (likelihood ratio test, adjusted p-value $< 0.01$, 7987 genes, S1 Table) were used as inputs for clustering.

## Gene Ontology enrichment

Gene Ontology (GO) annotations for the *S. mansoni* genes were downloaded from GeneDB. org. Identification of enriched GO terms (biological process terms) was performed using topGO (version 2.24.0) [32] with a *weight* algorithm and Fisher's exact test. All *S. mansoni* genes were used as a reference background for the enrichment analysis.

## Protein structure prediction

Amino acid sequences for proteins of interest were obtained from GeneDB.org. Protein 3D structures were predicted from amino acid sequences using I-TASSER online server (v5.0) [33–35] with default parameters. TM-scores indicate similarity between two structures. The value ranges between 0–1, with a value of one indicating a perfect match. Images of the predicted structures and their alignment were from.pdb files obtained from I-TASSER predictions or from Protein Data Bank (PDB) [36] and reproduced using Chimera software [37].

## Protein domain searching

Protein domains were identified from amino acid sequences using InterProScan online server (v60, v61 and v78) [38]. CathDB [39] was used to explore protein structural domains and to search by structural match.

# Results

## Parasite morphologies correlate with marked transcriptional signatures in developing parasites

Changes in the morphology and transcriptome during intra-mammalian development of *S. mansoni* were investigated in parasites collected from experimentally-infected mice at time points between 6–35 days post-infection (Fig 1).

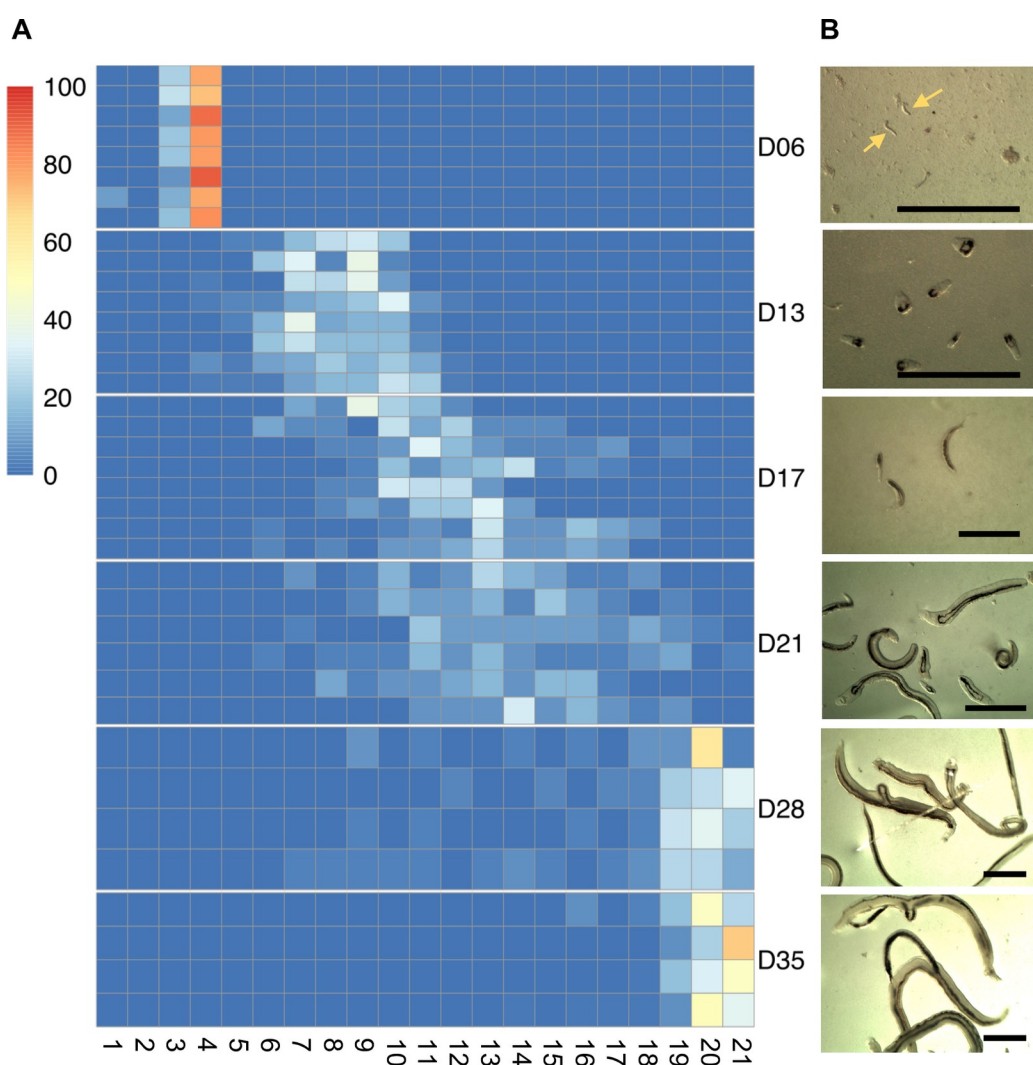

**Fig 2. Morphology of *in vivo S. mansoni*.** A) Morphological scoring of *S. mansoni* parasites collected at the indicated time points post-infection (D06 to D35). Heatmap columns represent the twenty-one distinct morphological groups following published scores [22], and heatmap rows indicate biological replicates of the infections, i.e. parasites collected from individual mice. Colours on the heatmap show the number of worms that fell into each morphological group, normalised as a percentage of the total number of worms that were scored within the replicate. B) Representative images of worms from each time point. Scale bars: 1 mm.

Lung schistosomula were morphologically homogeneous, whereas circulating larvae that had left the lungs (days 13 to 28) were heterogeneous in size and developmental progression (Fig 2), consistent with previous reports [3,22]. At 28 days post infection, most of the parasites had developed into adults and worm pairs started to become evident. All day-35 parasites were fully mature paired male and female adults (Fig 2).

Parasites from individual mice were pooled and each of these pools was considered a biological replicate. At least 3 biological replicates were obtained for each time point and changes in their transcriptomes were measured using RNA-seq. A principle components analysis showed tight clustering of biological replicates and a large variation among the time points (Fig 3). All replicates from lung schistosomula clustered separately from day-13 to day-21 groups, and the adult stages (days 28 and 35) clustered away from the other time points,

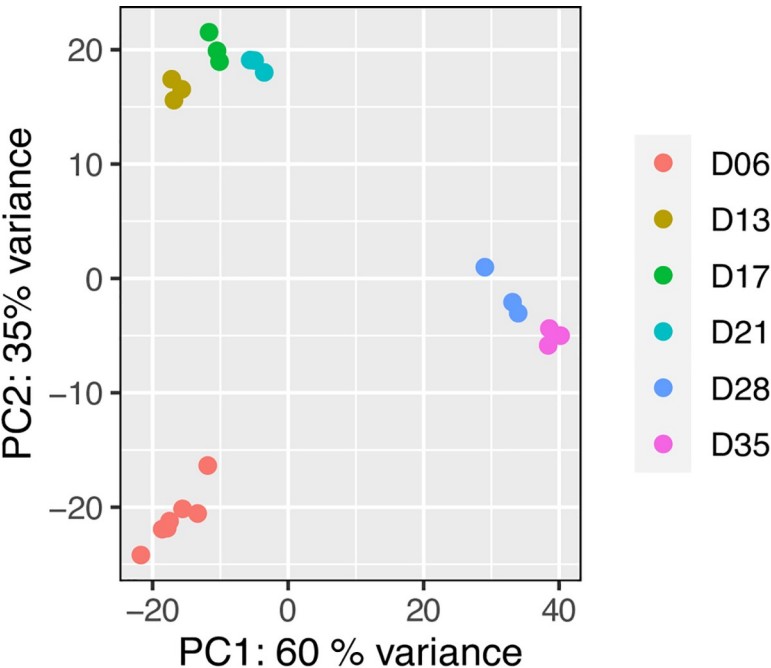

**Fig 3. Principal component analysis of the transcriptomic data from all time points.** PCA plot of all transcriptomic data based on rlog-transformed normalised read counts. Each dot represents the transcriptome from a pool of parasites collected from an individual mouse, i.e. one biological replicate.

indicating a good correlation between the morphological progress and marked transcriptional signatures of the developing parasite. The transcriptional differences observed between day-28 and day-35 parasites may be related to sexual maturation and egg laying, which is onset at day 35 [22]. In contrast, the transcriptional similarity of parasites at days 13, 17, and 21 may reflect genuinely similar gene expression profiles or the heterogeneity in the overlapping morphologies of parasites at those three time points.

## Gene expression changes associated to developmental milestones of *S. mansoni*

Comparing the transcriptomes of parasites collected 6- and 13-days post-infection provided information on the transition between lung schistosomula and circulating juveniles, i.e. parasites that have already left the lungs and have entered the systemic circulation *via* the pulmonary veins. In the lung schistosomula, 864 genes were up-regulated compared with day 13 juveniles (S2 Table). The up-regulated genes related to multiple signalling processes including signal transduction and neuronal signalling pathways (S3 Table). This suggests that neuronal activities were increased in the lung stage, compared with day-13 stage, perhaps for sensing and locomotion to allow parasite migration through the lung capillary bed, to reach the pulmonary veins and continue through the rest of bloodstream circulation [40]. In contrast, 686 genes were up-regulated in the day-13 schistosomula (S2 Table), including genes involved in mitosis and its associated processes, such as translation, post-translational modification, and transcriptional regulation (S3 Table). This is consistent with the growth phase described in day-13 worms and corroborates reports that mitosis is not detectable in the lung stage [3,41].

As the parasites develop from circulating juveniles to adult forms, up-regulated genes identified in the liver stage (day 21) compared to pre-egg-laying adult stage (day 28) were involved

in cell division, differentiation, and developmental regulation (S4 and S5 Tables). In contrast, gene expression in the day-28 worms showed a massive up-regulation of genes involved in egg production such as *tyrosinase* (Smp_050270, Smp_013540), *eggshell protein* (Smp_000430), and *Trematode Eggshell Synthesis domain containing protein* (Smp_077890) (S4 Table). The Gene Ontology (GO) terms *metabolic processes* and *biosynthesis processes* were enriched in up-regulated genes at day 28 (S5 Table), possibly also reflecting synthesis of compounds necessary for egg production (*tyrosinase* genes are annotated with the terms *organic hydroxy compound biosynthetic process*). However, the increased biosynthesis also likely reflected increased nutrient and energy requirements, and scavenging of host-derived substrates by the parasites because GO terms for lipid metabolic process, glycerol metabolic process, purine ribonucleoside salvage, and carbohydrate transport were also enriched (S5 Table). The genes contributing to the enriched GO term carbohydrate transport encode two confirmed glucose transporters (Smp_012440, Smp_105410) [42] and a third, non-confirmed, putative glucose transporter (Smp_139150).

Between days 28 and 35, the parasites become fully established in the portal system within the mesentery veins and lay large numbers of eggs. Expectedly, amongst the 72 genes that were up-regulated during the progression from day 28 to day 35, many were related to egg production (S6 Table). Proteases involved in blood feeding [43], *cathepsin B*, *D*, and *L* were also up-regulated (approximately 2.5-fold; S6 Table), consistent with the high nutrient requirement of egg-producing females. In addition, down regulation of genes involved in signalling and developmental control was evident (S6 and S7 Tables).

To further explore transcriptomic changes across all developmental stages analysed, genes were clustered into 96 groups based on their expression profile over the whole time-course (Fig 4 and S1 Fig). Specific expression patterns became evident for multiple clusters; for instance, clusters 1, 2, 9, 10, 17, 25, 26, 27, 33, and 34 showed increased expression in the developing parasites, from day 13 to day 28 post-infection (Fig 4 orange boxes and S1 Fig). These clusters comprised a total of 737 genes with the top five enriched GO terms related to cell replication and regulation (S8 and S9 Tables). Striking up-regulation in adult stages (day 28 and day 35) was seen in six clusters, particularly cluster 96 and, to a lesser extent, clusters 79, 80, 87, 88 and 95 (Fig 4 pink boxes, S1 Fig and S8 Table). These clusters comprised genes involved in egg production, such as two tyrosinase genes (Smp_013540 and Smp_050270) involved in protein cross-linking during egg-shell synthesis [44]. Two of the egg-shell precursors (Smp_131110 and Smp_014610) [45] were also present in cluster 96 (S8 Table). In addition, multiple genes of unknown function shared expression patterns with these genes related to egg-production, suggesting that they may have a related function or share the same pathway(s) (S8 Table).

## Signalling pathways, iron homeostasis and micro-exon genes (MEGs) in lung schistosomula

Given the scarcity of data on lung schistosomula, we investigated gene expression at this stage by performing pairwise comparisons between the lung stage and day-13 parasites, and by focusing on data that was clustered over multiple time points but showed striking differences in the lung. High expression in the lung stage, compared with other intra-mammalian stages, was seen mainly in three clusters (8, 24 and 32), where high expression on day 6 precipitously dropped to a low baseline for the rest of the time-course (Fig 4 yellow boxes, S1 Fig and S8 Table). The three clusters contained a total of 72 genes, and a GO term enrichment analysis suggested that many of these were involved in signalling, metabolism, transport and iron homeostasis (S10 Table).

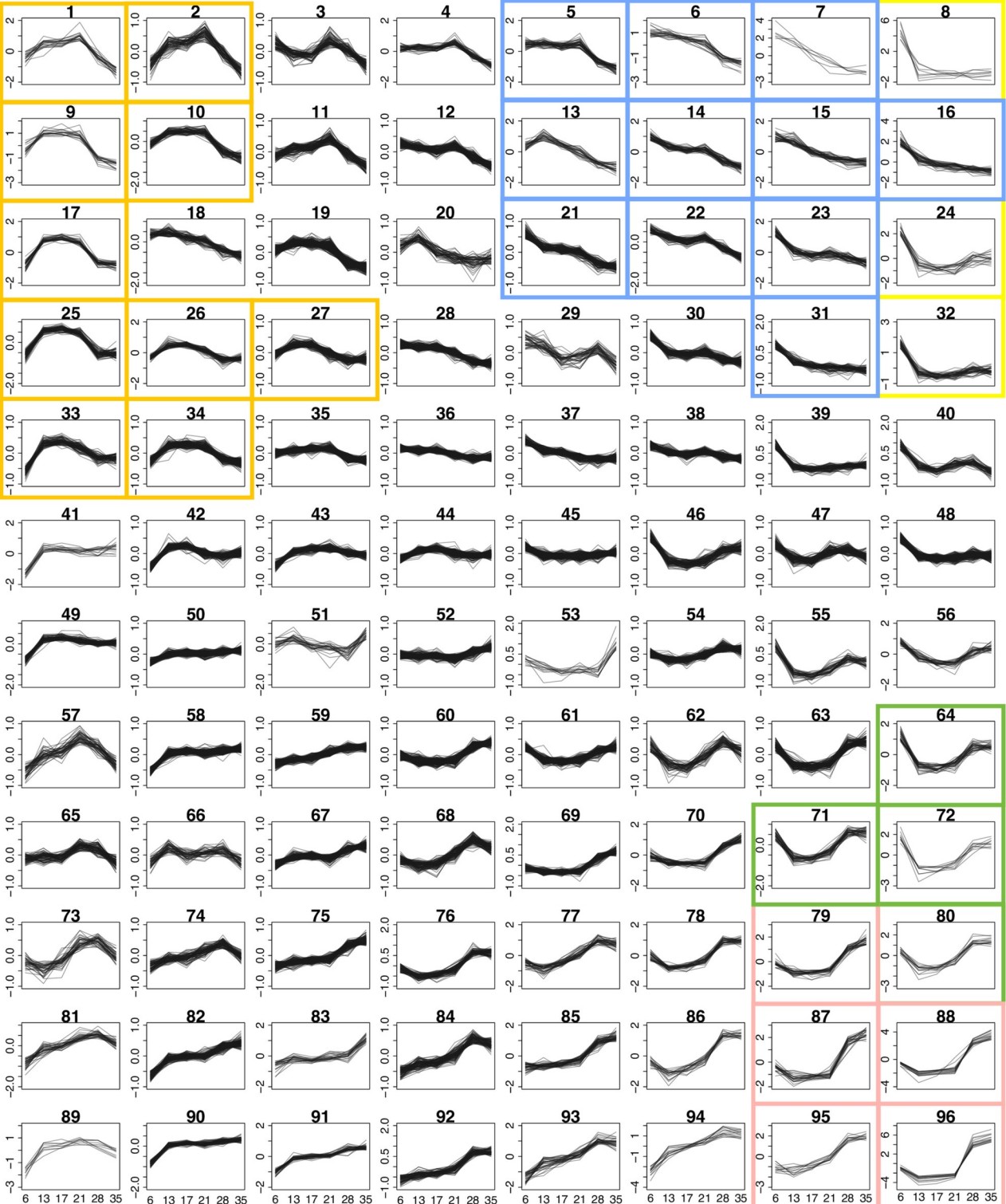

**Fig 4. Clusters of genes based on time-course expression pattern.** Expression profiles of genes showing differential expression in at least one time point, clustered into 96 groups. The clustering was based on mean-normalised rlog-transformed raw read counts over six time points. The y-axes are scaled independently to emphasise the differences between clusters. Plots with a single y-axis scale are shown in S1 Fig. The coloured boxes mark clusters that were part of the GO term enrichment analyses or were discussed in detail; orange, clusters of genes with increased expression during the liver stage; blue, high expression in the lung stage and steadily declined toward adult stages; yellow, high expression in the lung stage; green, high expression in the lung and adult stages with low expression during liver stages; pink, high expression in adult stages.

Genes related to developmental control were also over-represented in 11 clusters that showed high expression in lung stage followed by a steady decline towards adult (cluster 5, 6, 7, 13, 14, 15, 16, 21, 22, 23, 31; Fig 4 blue boxes, S1 Fig and S11 Table), including several transcription factors and cell adhesion proteins involved in embryogenesis and neuronal development, such as *SOX* (Smp_148110, Smp_161600) and *procadherin family* (such as Smp_011430, Smp_141740, and Smp_155600) [46,47]. In addition, *Wnt*, *nanos*, and *frizzled receptors*, important for cell-fate determination and control of development [48–50], showed a similar expression pattern.

Compared to day-13 parasites, the relative expression of multiple genes related to signalling processes was high in the lung stage; amongst 864 highly expressed genes, the top-four enriched GO terms were *signal transduction*, *male sex determination*, *potassium ion transport*, and *neurotransmitter transport* (S2 and S3 Tables). The enrichment of several other GO terms, albeit with weaker statistical support (p-values: 0.036–0.045), provided further evidence for the prominent role of signalling processes in the lung stage (S2 and S3 Tables). Several developmental terms were also enriched (e.g. *cell differentiation*, *homophilic cell adhesion*, *brain development*, *male sex determination*). The pronounced expression of genes involved in neuronal signalling–inferred from the GO terms *neuropeptide signalling pathway*, *sodium ion transport*, *chloride transport* and *neurotransmitter transport* (S2 and S3 Tables)–may reflect the parasite's responsiveness to environmental cues. Consistent with the broad picture provided by GO term enrichment, the top up-regulated genes in the lung stage included a *rhodopsin orphan GPCR* (Smp_203400), and two Ras related proteins (Smp_132500, Smp_125060), all of which were up-regulated more than 32-fold (Table 1).

The GO term *cellular iron ion homeostasis* was enriched amongst genes up-regulated in lung stage compared to day-13 schistosomula (S3 Table). For example, the *ferritin*-2 genes (Smp_047650, Smp_047660, Smp_047680; somal ferritin) displayed fold changes in the range 2.1–7.2 (S2 and S3 Tables). Although down-regulated in day-13 compared to lung stage schistosomula, these three *ferritin* genes were still expressed at a time when iron is required for growth and development (S2 Fig). Another ferritin isoform, ferritin-1 (Smp_087760; yolk ferritin), is associated with vitelline cells [51,52] and in the present study showed increasing expression as the parasite developed into adult stage (S2 Fig). However, sex-biased expression of ferritin-1 and 2 has previously been reported [51,52] and in our mixed-sex samples, we cannot rule out the possibility that our later collection time points contained a greater proportion of female worms. In contrast, other genes related to iron-sequestration were expressed at a similar level in the lung stage and day-13 stage, such as *putative ferric reductase* (Smp_136400) and *divalent metal transporter* (*DMT 1*) (Smp_099850) (S3 Fig). Both genes are hypothesised to be involved in the same pathway, with the *putative ferric reductase* cleaving iron from host transferrin (glycoprotein iron carrier) before transporting into the parasite *via* a *DMT* [53,54].

Micro-exon genes (MEGs), whose structures mainly comprise short exons with lengths that are multiples of three bases, are an abundant feature in parasitic helminths [55,56]. Despite expression across all developmental stages, and at least 41 MEGs being annotated and assigned into sequence-similarity families [55,57], little is known about their function. *MEG-14* has been shown to interact with an inflammatory-related human protein [58], and *MEG-3*, *MEG-14*, *MEG-15*, *MEG-17 and MEG-32*, were previously identified in the oesophagus of schistosomula or adults [59]. Multiple MEGs were up-regulated in the lung stage, e.g. 6 of the 72 genes in the major lung stage expression clusters (cluster 8, 24 and 32) were MEGs (S8 Table). Most strikingly, cluster 8 contains eight genes, of which four are micro-exon genes (MEGs); two from the *MEG-2* family and two from *MEG-3* family (S8 Table). Three of these MEGs (Smp_159810, Smp_138070, Smp_138080) were previously shown to be up-regulated in schistosomula, three days after *in vitro* transformation and detected in schistosomula or egg

**Table 1. Top 20 genes up-regulated in lung stage compared to day-13 schistosomula.**

| Gene identifier | *Log$_2$FC (lung/D13) | Adjusted p-value | Product name |
|---|---|---|---|
| Smp_138080 | 12.62 | 1.48E-19 | MEG-3 (Grail) family |
| Smp_138070 | 11.99 | 3.58E-30 | MEG-3 (Grail) family |
| Smp_159810 | 11.22 | 1.02E-48 | MEG-2 (ESP15) family |
| Smp_159800 | 9.66 | 5.10E-28 | MEG-2 (ESP15) family |
| Smp_181510 | 9.58 | 1.15E-13 | hypothetical protein |
| Smp_032990 | 8.98 | 7.14E-13 | Calmodulin 4 (Calcium binding protein Dd112) |
| Smp_159830 | 8.69 | 1.00E-05 | MEG-2 (ESP15) family |
| Smp_138060 | 8.06 | 3.76E-09 | MEG-3 (Grail) family |
| Smp_203400 | 7.34 | 1.88E-07 | rhodopsin orphan GPCR |
| Smp_005470 | 7.31 | 3.34E-08 | dynein light chain |
| Smp_077610 | 6.61 | 6.19E-07 | hypothetical protein |
| Smp_166350 | 6.59 | 1.31E-06 | hypothetical protein |
| Smp_180330 | 5.97 | 2.91E-32 | MEG 2 (ESP15) family |
| Smp_205660 | 5.84 | 4.52E-15 | hypothetical protein |
| Smp_033250 | 5.80 | 4.43E-05 | hypothetical protein |
| Smp_132500 | 5.73 | 4.50E-04 | ras and EF hand domain containing protein |
| Smp_152730 | 5.68 | 3.41E-11 | histone lysine N methyltransferase MLL3 |
| Smp_241430 | 5.61 | 3.89E-23 | Aquaporin 12A |
| Smp_125060 | 5.55 | 6.70E-04 | kinase suppressor of Ras (KSR) |
| Smp_198060 | 5.51 | 4.73E-13 | hypothetical protein |

*Log$_2$FC (lung/D13), logarithm of the fold change in expression level between lung stage and day-13 schistosomula

secretions [57]. Furthermore, a pairwise comparison revealed a total of 17 MEGs, from seven families, that were up-regulated in lung stage compared to day-13 schistosomula (S2 Table), with seven MEGs amongst the top 20 lung-stage up-regulated genes (Table 1). In contrast, only one MEG, a member of *MEG-32*, was up-regulated in day-13 schistosomula compared to the lung stage (S2 Table).

## Lung stage expressed genes with anti-inflammatory roles

Genes involved in defence against oxidative stress were highly expressed in lung schistosomula, presumably to neutralise reactive oxygen species (ROS) produced during inflammation. For instance, expression of *extracellular superoxide dismutase* was 17-fold higher than in day-13 parasites (Smp_174810) (S2 Table and S4 Fig). Superoxide dismutase catalyses the detoxification of superoxide into hydrogen peroxide and molecular oxygen [54,60]. The antioxidant *thioredoxin peroxidase* (Smp_059480) that neutralises peroxide [61] was similarly up-regulated (more than 16-fold) in the lung stage (S2 Table) and thioredoxin glutathione reductase (TGR; Smp_048430), another important redox enzyme [62], showed marginally higher expression in the lung stage (S5 Fig). The gene encoding *single Kunitz protease inhibitor* (Smp_147730, S6 Fig), putatively involved in host immune defence [63], was also highly expressed in the lung stage (16-fold greater than day 13, adjusted p-value < 1E-100) (S2 Table).

These three genes with particularly striking up-regulation in the lung stage belong to the same cluster (cluster 72). Given the possible roles of cluster 72 genes in counteracting oxidative stress and in host immune system evasion, other genes from this cluster were explored in more detail. Cluster 72 contained seven additional genes (Table 2). Three encoded hypothetical proteins of unknown function, while the other four were predicted, based on sequence

**Table 2. Genes in cluster 72.**

| Gene identifier | Product name |
|---|---|
| Smp_059480 | thioredoxin peroxidase |
| Smp_059980 | arginase |
| Smp_074570 | hypothetical protein |
| Smp_114660 | hypothetical protein |
| Smp_134870 | early growth response protein |
| Smp_147730 | single Kunitz protease inhibitor; serine type protease inhibitor |
| Smp_156510 | PDZ and LIM domain protein 7 |
| Smp_166920 | PDZ and LIM domain protein Zasp |
| Smp_174810 | Extracellular superoxide dismutase (Cu Zn) |
| Smp_182770 | hypothetical protein |

similarity, to encode proteins with recognisable products. The first, *arginase* (Smp_059980), is hypothesised to counteract the host immune response by depleting L-arginine from blood, thereby preventing it from being used by macrophages in the production of nitric oxide [64,65]. The second, a schistosome homologue of *early growth response protein* (EGR, Smp_134870) displayed zinc finger and DNA-binding domains suggesting a transcription factor role, although its targets are not known. The third and fourth proteins contained *PDZ* and *LIM* domains (Smp_156510, Smp_166920) but these also have no obvious link with host immune evasion.

Cluster 64 contained genes with similar expression profiles to cluster 72 and like the latter cluster, was annotated with GO terms enriched for redox processes (S12 Table). Clusters 64 and 72 also contained genes involved in cellular metabolism and transport. However, enrichment for these specific GO terms became more pronounced when two additional clusters (71 and 80), with reduced lung stage expression (Fig 4 green boxes), were included in the analysis (S13 and S14 Tables). Amongst the top enriched GO terms for all four clusters was *carbohydrate transport* with two glucose transporters (Smp_012440 and Smp_105410) that have been previously characterised in *S. mansoni* [42].

## Hypothetical protein with predicted structure matching a complement cascade regulator

With four genes out of ten in cluster 72 potentially involved in host immune interactions, the three that encoded hypothetical proteins (Smp_074570, Smp_114660, Smp_182770) were investigated further. Peptides encoded by Smp_074570 are abundant (top 10) in secreted extracellular vesicles produced by *S. mansoni* schistosomula [66], suggesting a role in host-parasite interactions. No further published information was found for products of the other two genes, and both contained no known signature domains. I-TASSER protein-structure prediction software [33–35] was therefore used to align predicted structures against known structures in the Protein Data Bank (PDB) [36]. No match was found for Smp_074570 and Smp_114660 but, remarkably, Smp_182770 produced a single high confidence match (TM-score > 0.9) to human complement factor H (CFH; PDB entry 3GAW) [67] (Fig 5).

Homologues of the Smp_182770 are present in other *Schistosoma* species as well as other flatworms, such as the liver flukes *Fasciola*, *Opisthorchis*, and *Clonorchis*, and the intestinal fluke *Echinostoma* (Fig 6A), suggesting an evolutionarily-conserved role [71,72]. The Smp_182770 locus in *S. mansoni* genome is immediately adjacent to Smp_038730, another hypothetical protein. RNA-seq mapping suggested that the two were in fact parts of the same

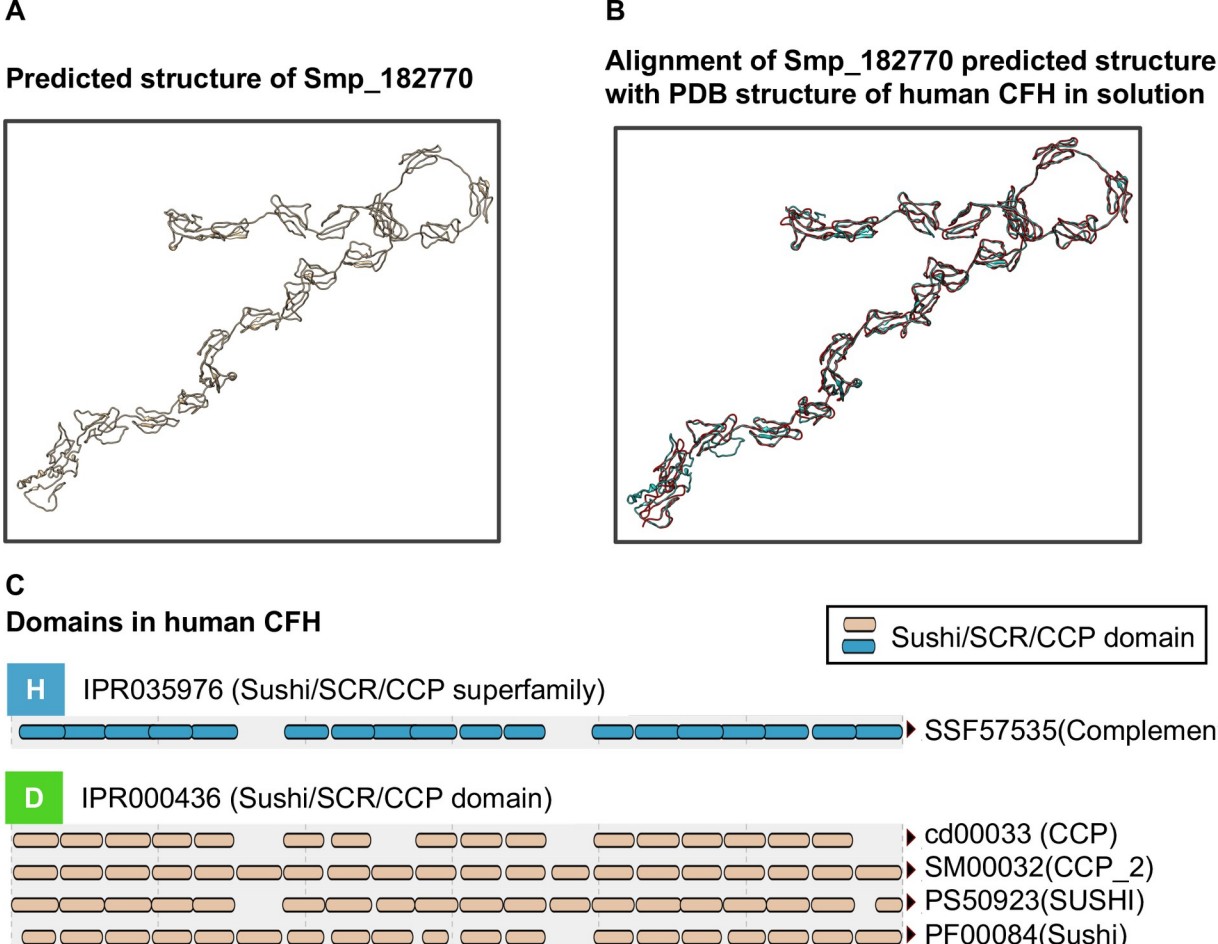

**Fig 5. Predicted structure of Smp_182770 aligned with structure of human CFH.** (A) Three-dimensional structure of Smp_182770 predicted using the I-TASSER server. (B) Alignment between the predicted structure (blue) and 3D structure of human CFH (red) in 250 mM NaCl buffer, obtained from PDB (entry 3GAW). (C) Domain components of human CFH identified from multiple databases using the InterProScan web server [38]. SCR, short consensus repeat; CCP, complement control protein. Both SCR and CCP are alternative names of the *sushi* domain. CFH is a well-characterised regulator of the complement cascade that is found on the surface of human cells and prevents complement attack on self [68,69]. The human CFH (PDB entry 3GAW) contains *sushi/ccp* domain repeats (Fig 5C) involved in regulating complement cascade. Smp_182770 does not contain the *sushi/ccp* domain repeats but it does encode tandem repeats similar to those expected from mammalian CFH genes [70].

gene (Fig 6B). From gene clustering, Smp_038730 is in cluster 24 which has a similar expression pattern to cluster 72, but with noticeably lower expression in the adult stage (Fig 4 yellow box and S1 Fig). In a subsequent version of the *S. mansoni* genome (available from parasite. wormbase.org), the two genes have been merged to become Smp_334090. The predicted structure of Smp_334090 also resembles that of human CFH (TM-score = 0.830) (S7 Fig).

## Discussion

We have described a complete transcriptome time-course of *S. mansoni* covering key developmental stages during the intra-mammalian infection, including the first RNA-seq transcriptome of *in vivo* lung schistosomula. The data recapitulated known major biological changes for well-characterised parts of the life cycle in the mammalian host, as well as provided novel insights into molecular processes underlying parasite development and interactions with the

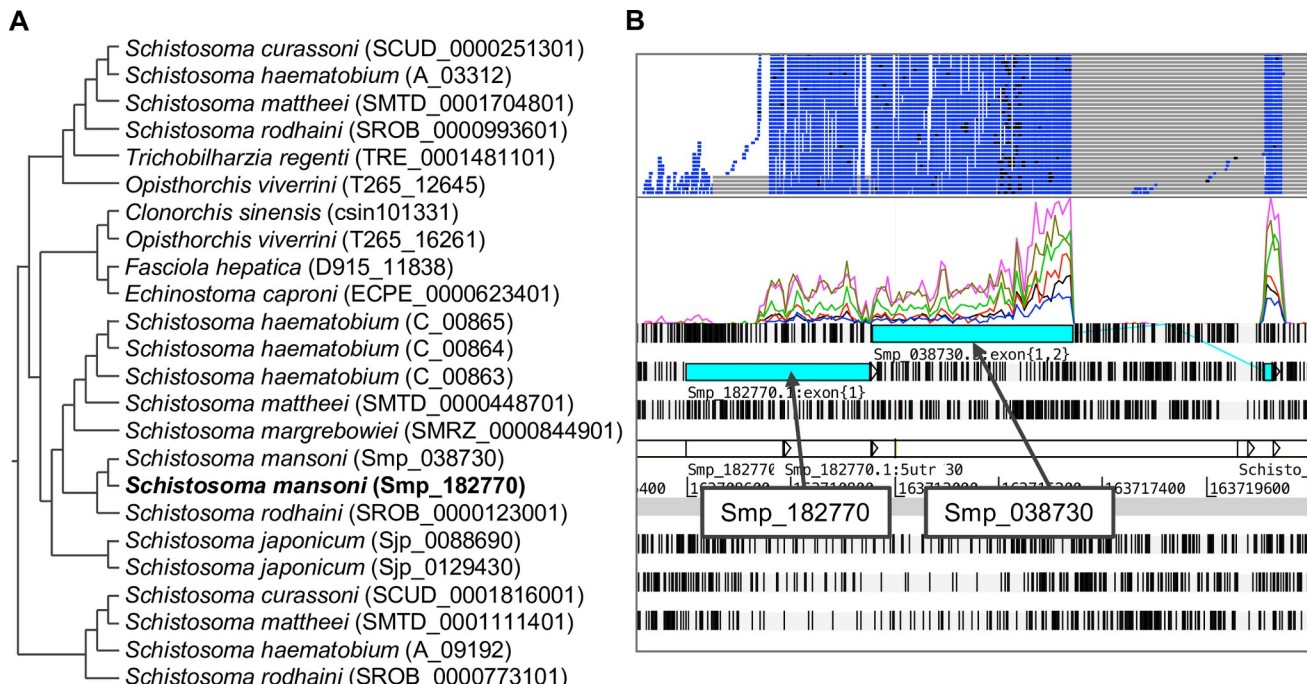

**Fig 6. Homologous relationship of Smp_182770 and alignment of RNA-seq reads to genomic locations.** A) Homologues of Smp_182770. Information on homologues of genes was obtained from WormBase ParaSite, release 8 [23], based on gene-trees generated by the Compara pipeline [73]. Gene identifiers from WormBase ParaSite are shown in parentheses after the species names. B) Artemis screenshot showing the genomic region that contains Smp_182770 and RNA-seq reads mapped from multiple RNA-seq libraries (two top rows).

host. In particular, analysis of the lung developmental stage highlighted striking signalling pathways, including those related to developmental control, cell differentiation and neuropeptide signalling. In addition, tentative strategies for immune evasion such as up-regulation of iron homeostasis, oxidative stress-related genes, and anti-inflammatory genes were revealed.

In early stages of *S. mansoni* infection, schistosomula elongate and migrate within lung capillaries [74]. Although the cues involved in these changes are unknown [75], it is believed that schistosomula differentiate or remodel their existing cells, rather than trigger cell division [3,41]. The up-regulation of signalling-related genes in this developmental stage emphasises processes required for successful migration through lung capillaries. Furthermore, changes in development-related genes may play roles in the remodelling or trigger developmental onset observed in later stages. In some parasitic trematodes, fixed behavioural patterns have been associated with specific developmental stages involved in processes such as migration, site-finding, and adhesion [76,77]. Given that the lung stage undergoes major morphological changes, an increase in neuronal activities indicated by our data may reflect distinct behaviours associated with the new phase of development, as well as responses to a new environment within the lung capillaries. Neuronal signalling is one of the processes disturbed in irradiated schistosomes [78] many of which exit lung capillaries into the alveoli and eventually die during lung migration [20,74,79]. The migration through the lung capillaries represents a challenge for the schistosomula. Moving within narrow lung capillaries, the schistosomula are in direct contact with the capillary endothelium [74] and they may interrupt blood flow, increasing the risk of microthrombosis or leading to an accumulation of circulating immune cells [80]. However, it has been shown that lung schistosomula can resist direct cytotoxic immune killing [19,79,81,82]. Lung inflammation is observed during the infection, but it appears not to be

associated with the location of migrating parasites, but instead triggered by damaged host tissue [83]. The accumulation of immune cells, instead of killing the parasites, is thought to either disrupt blood vessels, inducing microhemorrhage and causing the parasites to migrate into the alveoli, or it may act as a plug that blocks migration [20,84].

Our data revealed that multiple genes with potential immune evasion or protective roles are up-regulated. One example was the *Kunitz protease inhibitor* that has an anti-coagulation and anti-inflammation role [63] and has been proposed as a vaccine candidate against *S. mansoni* [85]. Neutralisation of oxidative stress by up-regulating dismutase and peroxidase systems may mitigate the effects of inflammation caused by migrating schistosomula. Both reactive oxygen species (ROS) and iron metabolism are involved in the host immune response; therefore, iron homeostasis and ROS neutralisation have likely evolved as an immune evasion strategy in these parasites [54]. Schistosomula also clearly face the risk of oxidative damage from the uniquely high oxygen tension in the lung [86,87] and therefore have a further requirement for maintaining redox balance. In rodents, expression of thioredoxin is induced by oxidative stress [87]. The up-regulation of schistosome ROS metabolism likely also depends on changes in the environment rather than a hard-wired gene expression programme. In contrast to our *in vivo* results from lung schistosomula, genes related to stress and immune evasion are expressed at a lower level when the parasites are cultivated *in vitro* [8], highlighting the need to study parasites obtained from their most natural environment to understand relevant biological processes.

Multiple genes with tentative roles in immune evasion displayed particularly high expression in the lung stage, moderately high expression in adult stages and lower expression in the liver stages. In particular, a previously uncharacterised gene, annotated as a hypothetical protein, with this expression pattern was predicted to be structurally similar to the human complement factor H (CFH), a regulator of the complement cascade. In mammals, CFH cleaves C3b, a central protein in the complement cascade [88]. The regulatory function of CFH involves two other molecules—its cofactor complement factor I (CFI) [69] and decay-accelerating factor (DAF) [68]. To date, a *S. mansoni* gene with sequence or structural similarity to CFI has not been described, but the parasites do possess a serine protease, m28, that cleaves C3bi (a function normally carried out by CFI [68,89]). For DAF, a native *S. mansoni* form has not been reported but schistosomes can acquire DAF from host blood cells [90,91]. Given the nature of other genes sharing its expression pattern, the structural match to human CFH, and its phylogenetic relationship, it is tempting to speculate that the CFH-like gene in schistosomes may have a role in immunomodulation. Nevertheless, host immune system proteins have been shown to interact with schistosomes in their developmental signalling pathways [92]. CFH in mammals not only cleaves C3b, but also has been implicated in lymphocyte extracellular secretion and DNA synthesis [92]. In addition, schistosomes have a 130 kDa surface protein that can bind to C3, inactivating the C3 complement-activation, and the binding stimulates renewal of the parasite surface membrane [88,93]. To have another, CFH-like protein identified here, inhibiting the downstream product of C3 activation, may seem redundant. However, its potential roles in immunomodulation should not be ruled out. It is common for pathways to have regulators at various steps, particularly for C3 which is focal for complement cascade activation and is involved in clearing *S. mansoni* after praziquantel administration [88,94].

Certain limitations need to be borne in mind when interpreting the data from this study. First, it has been shown that gene expression can change with the types of hosts [95,96] and, therefore, this limits extrapolation of findings from mouse host to humans. In *S. japonicum*, microarray data of adult worms collected from two different mouse strains, rabbits and water buffaloes (natural host) displayed highly correlated expression profiles across different host types. Nevertheless, over 400 genes were identified as differentially expressed, particularly

including those likely to be involved in signal transduction and metabolism [95]. Second, our transcriptome data were produced by 'bulk' RNA-seq from pooled mixed-sex worms. In bulk RNA-seq, strong signals from rarer cells types can be averaged out, and gene expression changes can therefore be missed if, for instance, a gene is up-regulated in one cell type but down-regulated in another. Single-cell transcriptomic analyses promise to overcome this limitation [97]. Using mixed-sex worms in our study was another possible confounder for genes, such as ferritin-1, where there is differential expression between male and female worms. However, the majority of our analyses focused on earlier stages where the sexes show little or no differentiation. Despite these various caveats that are common to many bulk analyses, the good agreement with key known events during parasite development provides confidence that the major transcriptional signatures we have detected reflect major changes within the parasite.

The data produced from this study will serve as a unique resource for the research community to explore changes across intra-mammalian stages of schistosome development. Our particular focus on the lung stage demonstrated consistency with previous observations and introduced potential new players in host-parasite interactions and parasite development. Further investigation and functional validation of genes identified here will help to decipher mechanisms for parasite long-term survival within the mammalian host, exposing vulnerabilities that can be exploited to develop new control strategies for this neglected tropical pathogen.

## Supporting information

**S1 Fig. Clusters of genes based on timecourse expression pattern, with fixed y-axes.** Expression profile of genes differentially expressed in at least one time point clustered into 96 groups. The clustering was done on mean-normalised regularized log transformation (rlog- transformed) of raw read counts. X-axes represent six time points from this dataset; y-axes represent the mean-normalised rlog-transformed. Unlike Fig 4, this supplementary figure shows clusters with a fixed range for the y-axis across all clusters in order to visualise clusters with the largest changes over time.
(TIF)

**S2 Fig. Expression of genes encoding ferritin-heavy chain over the timecourse.** Each dot represents one replicate from each of the time points. Y-axis represents normalised counts from DESeq2.
(TIF)

**S3 Fig. Expression of genes encoding putative ferric reductase (cytochrome b 561; Smp_136400) and Divalent Metal Transporter (DMT1) (Smp_099850) over the timecourse.** Each dot represents one replicate from each of the time points. Y-axis represents normalised counts from DESeq2.
(TIF)

**S4 Fig. Expression of genes encoding two extracellular superoxide dismutases (Smp_174810 and Smp_095980) over the timecourse.** Expression of Smp_174810 and Smp_095980 with y axes on log scale. Each dot represents one replicate from each of the time points. Y-axis represents normalised counts from DESeq2. Smp_095980 was identified as differentially expressed (S2 Table and S4 Fig), but this figure shows that its expression in the lung stage was high in only one out of seven replicates.
(TIF)

**S5 Fig. Expression of genes encoding thioredoxin glutathione reductase over the time-course.** Each dot represents one replicate from each of the time points. Y-axis represents normalised counts from DESeq2. Log$_2$FC between D13/D06 is -0.52, adjusted p-value for differential expression between D13/D06 is 4.10e-21.
(TIF)

**S6 Fig. Expression of genes encoding single Kunitz serine protease inhibitor over the time-course.** Each dot represents one replicate from each of the time points. Y-axis represents normalised counts from DESeq2.
(TIF)

**S7 Fig. Predicted structure of Smp_334090 aligned with structure of human CFH.** A) Predicted 3D structure of Smp_334090 (resulting from a merge of Smp_182770 and Smp_038730 in the most recent version of the *S. mansoni* genome) by I-TASSER based on the amino acid sequence. B) Alignment between the predicted structure (blue) and 3D structure of human CFH (in 137 mM NaCl buffer) obtained from PDB (PDB identifier: 3GAV) (red).
(TIF)

**S1 Table. Genes that were differentially expressed in at least one time point (likelihood ratio test, adjusted p-value $<$ 0.01).**
(XLSX)

**S2 Table. Differentially expressed genes between day-13 and lung stage *S. mansoni*.**
(XLSX)

**S3 Table. Enriched GO terms of genes differentially expressed between day-13 and lung stage *S. mansoni*.**
(XLSX)

**S4 Table. Differentially expressed genes between day-28 and day-21 *S. mansoni*.**
(XLSX)

**S5 Table. Enriched GO terms of genes differentially expressed between day-28 and day-21 *S. mansoni*.**
(XLSX)

**S6 Table. Differentially expressed genes between day-35 and day-28 *S. mansoni*.**
(XLSX)

**S7 Table. Enriched GO terms of genes differentially expressed between day-35 and day-28 *S. mansoni*.**
(XLSX)

**S8 Table. Genes differentially expressed in at least one time point identified by their clustered expression profiles.**
(XLSX)

**S9 Table. GO term enrichment of genes with high expression during liver stages (genes in cluster 1, 2, 9, 10, 17, 25, 26, 27, 33, 34).**
(XLSX)

**S10 Table. GO enrichment of genes up-regulated in lung stage (genes in cluster 8, 24, 32).**
(XLSX)

**S11 Table. Enriched GO terms of genes with high expression in lung stage followed by a steady decline toward adult (genes in cluster 5, 6, 7, 13, 14, 15, 16, 21, 22, 23, 31).**
(XLSX)

**S12 Table. Enriched GO terms of genes with high expression in lung stage, low expression in liver stages, and increased expression in adult stages (genes in cluster 64, 72).**
(XLSX)

**S13 Table. Enriched GO terms of genes with high expression in lung stage, low expression in liver stages, and increased expression in adult stages (genes in cluster 71, 80).**
(XLSX)

**S14 Table. Enriched GO terms of genes with high expression in lung stage, low expression in liver stages, and increased expression in adult stages (genes in cluster 64, 71, 72, 80).**
(XLSX)

**S1 File. RNA-seq details.** Total number of reads and % mapped to *S. mansoni* genome for each sample.
(DOCX)

**S2 File. Genome annnotation file.** GFF file containing annotation for *S. mansoni* longest transcript. The file was used for generating read counts from HTSeq-count.
(GFF)

**S3 File. Final counts.** A zipped folder containing final counts from HTSeq-count. The folder contains the following files: D06_SM_1_17675_4_1.htseq-count.txt, D06_SM_2_17675_4_2.htseq-count.txt, D06_SM_3_17675_4_3.htseq-count.txt, D06_SM_4_17675_4_4.htseq-count.txt, D06_SM_5_17675_4_5.htseq-count.txt, D06_SM_6_17675_4_6.htseq-count.txt, D06_SM_7_17675_4_7.htseq-count.txt, D13_SM_1_17675_4_8.htseq-count.txt, D13_SM_2_17675_4_9.htseq-count.txt, D13_SM_3_17675_4_10.htseq-count.txt, D17_SM_1_17675_4_11.htseq-count.txt, D17_SM_2_17675_4_12.htseq-count.txt, D17_SM_3_17675_4_13.htseq-count.txt, D21_SM_1_17675_4_14.htseq-count.txt, D21_SM_2_17675_4_15.htseq-count.txt, D21_SM_3_17675_4_16.htseq-count.txt, D28_SM_1_17675_4_17.htseq-count.txt, D28_SM_2_17675_4_18.htseq-count.txt, D28_SM_3_17675_4_19.htseq-count.txt, D35_SM_1_17675_4_20.htseq-count.txt, D35_SM_2_17675_4_21.htseq-count.txt, D35_SM_3_17675_4_22.htseq-count.txt.
(ZIP)

## Acknowledgments

We thank Prof Karl Hoffmann and Dr Cinzia Cantacessi for their comments on the study and the first version of this manuscript. We thank multiple members of the Parasite Genomics team at the Wellcome Sanger Institute for their comments and input for the experimental design and analysis; in particular, we thank Hayley Bennett, Lia Chappell, James Cotton, Stephen Doyle, Magda Lotkowska, Thomas Otto, Kate Rawlinson, Adam Reid, Alan Tracey and Gavin Rutledge. The infrastructure used for the analysis is maintained by the core IT Service and the Pathogen Informatics teams at the Wellcome Sanger Institute.

## Author Contributions

**Conceptualization:** Arporn Wangwiwatsin, Anna V. Protasio, Shona Wilson, Gabriel Rinaldi, Matthew Berriman.

**Data curation:** Arporn Wangwiwatsin.

**Formal analysis:** Arporn Wangwiwatsin.

**Funding acquisition:** Matthew Berriman.

**Investigation:** Arporn Wangwiwatsin, Anna V. Protasio, Christian Owusu, Mike J. Doenhoff, Gabriel Rinaldi.

**Methodology:** Arporn Wangwiwatsin, Anna V. Protasio, Mike J. Doenhoff, Gabriel Rinaldi, Matthew Berriman.

**Project administration:** Nancy E. Holroyd, Mandy J. Sanders.

**Resources:** Jacqueline Keane, Mike J. Doenhoff, Matthew Berriman.

**Software:** Arporn Wangwiwatsin, Anna V. Protasio, Jacqueline Keane.

**Supervision:** Anna V. Protasio, Shona Wilson, Gabriel Rinaldi, Matthew Berriman.

**Validation:** Arporn Wangwiwatsin.

**Visualization:** Arporn Wangwiwatsin.

**Writing – original draft:** Arporn Wangwiwatsin, Matthew Berriman.

**Writing – review & editing:** Arporn Wangwiwatsin, Anna V. Protasio, Mike J. Doenhoff, Gabriel Rinaldi, Matthew Berriman.

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
