## [Decision Letter · Decision Letter 0]

4 Nov 2019

Dear Dr Berriman:

Thank you very much for submitting your manuscript "Transcriptome of the parasitic flatworm Schistosoma mansoni during intra-mammalian development" (PNTD-D-19-01474) for review by PLOS Neglected Tropical Diseases. Your manuscript was fully evaluated at the editorial level and by independent peer reviewers. The reviewers appreciated the attention to an important topic but identified a few aspects of the manuscript that should be improved.

We therefore ask you to modify the manuscript according to the review recommendations before we can consider your manuscript for acceptance. Your revisions should address the specific points made by each reviewer.

(1) A letter containing a detailed list of your responses to the review comments and a description of the changes you have made in the manuscript.

(2) Two versions of the manuscript: one with either highlights or tracked changes denoting where the text has been changed (uploaded as a "Revised Article with Changes Highlighted" file ); the other a clean version (uploaded as the article file).

(3) If available, a striking still image (a new image if one is available or an existing one from within your manuscript). If your manuscript is accepted for publication, this image may be featured on our website. Images should ideally be high resolution, eye-catching, single panel images; where one is available, please use 'add file' at the time of resubmission and select 'striking image' as the file type. 

Please provide a short caption, including credits, uploaded as a separate "Other" file. If your image is from someone other than yourself, please ensure that the artist has read and agreed to the terms and conditions of the Creative Commons Attribution License at http://journals.plos.org/plosntds/s/content-license (NOTE: we cannot publish copyrighted images). 

(4) Appropriate Figure Files 

Please remove all name and figure # text from your figure files upon submitting your revision. Please also take this time to check that your figures are of high resolution, which will improve both the editorial review process and help expedite your manuscript's publication should it be accepted. Please note that figures must have been originally created at 300dpi or higher. Do not manually increase the resolution of your files. For instructions on how to properly obtain high quality images, please review our Figure Guidelines, with examples at: http://journals.plos.org/plosntds/s/figures

While revising your submission, please upload your figure files to the Preflight Analysis and Conversion Engine (PACE) digital diagnostic tool, https://pacev2.apexcovantage.com/ PACE helps ensure that figures meet PLOS requirements. To use PACE, you must first register as a user. Then, login and navigate to the UPLOAD tab, where you will find detailed instructions on how to use the tool. If you encounter any issues or have any questions when using PACE, please email us at figures@plos.org.

We hope to receive your revised manuscript by November 24, 2019. If you anticipate any delay in its return, we ask that you let us know the expected resubmission date by replying to this email.

To submit your revised files, please log in to https://www.editorialmanager.com/pntd/

Sincerely,

Matty Knight, Ph.D

Associate Editor

Walderez Dutra

Deputy Editor

Reviewer's Responses to Questions

**Key Review Criteria Required for Acceptance?**

**Methods**

-Are the objectives of the study clearly articulated with a clear testable hypothesis stated?

-Is the study design appropriate to address the stated objectives?

-Is the population clearly described and appropriate for the hypothesis being tested?

-Is the sample size sufficient to ensure adequate power to address the hypothesis being tested?

-Were correct statistical analysis used to support conclusions?

-Are there concerns about ethical or regulatory requirements being met?

Reviewer #1: See below

Reviewer #2: The objectives of the study are clear and study design is appropriate as presented, but I am concerned that previous knowledge from studies of additional life cycle stages are not discussed.

The sample size is adequate and there are no ethical concerns.

Reviewer #3: The authors’ submitted manuscript “Transcriptome of the parasitic flatworm Schistosoma mansoni during intra-mammalian development” investigated the temporal changes in the gene expression of S. mansoni in mice. This work has highlighted gaps in current understanding of S. mansoni infection and has provided valuable data adding to the current knowledge of the possible molecular mediators during host-parasite infection by studying gene expression of six developmental stages of the parasite. The authors used RNA-seq to analyze the transcriptomic profiles of the parasites and used bioinformatics tools to further analyze the biological significance of the gene expression. The authors have also done a thorough analysis and speculating signaling pathways that might be involved during the infection based on the expression profiles of genes showing differential expression at different developmental stages by performing cluster and GO term analysis. The authors further investigated the three hypothetical proteins that are involved in host immune interactions by protein structure prediction. Based on differential expression analysis the authors found micro-exon-genes (MEGs) to be highly up-regulated in day 6 as compared to day-16 schistosomula recapitulated some of the data shown in previous studies. In addition, data from the manuscript showed that the major signaling pathways regulated during these six developmental stages are those related to developmental control, cell differentiation and host interaction (including oxidative stress, iron homeostasis and inflammation) and found that one of the proteins aligned with the structure of human CFH, a regulator of the complement cascade. Overall this manuscript is clearly written and easy to follow. Methods used to generate the transcriptome data were also well presented. 

Minor comments:

1. (Line 77) Previous studies on intra-mammalian development of S. mansoni were mentioned and it would have been helpful for the general audience to include statements on the major findings/summaries from those in vitro/in vivo studies. 

2. Authors should include details of the raw RNA-seq data including number of reads obtained and % of mapping.

**Results**

-Does the analysis presented match the analysis plan?

-Are the results clearly and completely presented?

-Are the figures (Tables, Images) of sufficient quality for clarity?

Reviewer #1: See below

Reviewer #2: The analyses presented match the study plan and the results are clearly and completely described and the figures are clear.

Reviewer #3: see above

**Conclusions**

-Are the conclusions supported by the data presented?

-Are the limitations of analysis clearly described?

-Do the authors discuss how these data can be helpful to advance our understanding of the topic under study?

-Is public health relevance addressed?

Reviewer #1: See below

Reviewer #2: The conclusions are supported but the limitations could be better addressed. The authors clearly describe how the results can advance the field.

Reviewer #3: see above

**Editorial and Data Presentation Modifications?**

Reviewer #1: See below

Reviewer #2: None

Reviewer #3: see above

**Summary and General Comments**

Reviewer #1: Although the paper and approach are not entirely novel, the work presented herein presents a new and authoritative understanding of gene expression during development of Schistosome mansoni in its experimental host. The work uncovers the profiles of molecules involved in a range of activities at the host-parasite interface. The separation of lung-phase worms appears to be quite rapid compared with those of previous studies and there is some confidence that the transcriptome(s) of this stages approaches the native state. 

The manuscript is clear and well presented and will make a fine contribution. Some minor comments are made below to assist improve an already strong manuscript.

1. Line 262. Upregulation of molecules associated with neuronal function. One very interesting analysis of Fasciola development in its mammal host by MV Sukhdeo of Rutgers University suggested that trematodes followed fixed actions patterns of behaviour, that is, the parasites displayed certain behaviours that facilitated their development to a certain point. Having reached that point, a whole new set of behaviours were initiated. The lung stage of Schistosoma seems critical in development and host-parasite interaction and some major morphological changes occur in the parasites in the time they are present in the lungs. So, just as a thought exercise, I wonder if it is worthwhile adding that the neuronal activity is also associated with a transition to a whole new phase of development? 

2. I was happy to see reference to iron metabolism and the putative iron transporters and related genes. The findings of ferritin are interesting. Work on S mansoni (Schüssler et al Mol Reprod Dev. 1995 Jul;41(3):325-30) and S japonicum (Jones 2007;39(9):1646-58) demonstrate that one form of ferritin (called yolk ferritin) is associated with vitelline cells. Iron was proposed to be either stored in vitelline cells as part of the requirements for development in the snail (Schüssler) or to stabilise the egg shells (Jones). One would expect or hope to see one ferritin shoot up in its transcriptomic profiles in older worms. Is there any evidence of this occurring. Given the statement above, does the drop in ferritin levels at day 13 seem somewhat strange?

3. I may have missed it, but do the authors have a way of determining the numbers of males and females in each host at different stage? Is the gender balance the same in all mice? Implicit in this question is the concept that males and females will, or be expected to, have distinct transcriptional profiles at all stages. Would a difference in gender balance between mice or between stages skew the transcriptomes? Thus, is there a way of, using either male-only or female-only genes as markers of the numbers of respective gender balance?

Reviewer #2: The research presented here in the manuscript of Wangwiwatsin et al, is experimentally sound and examines difficult to study in vivo lifecycle stages with the power of and quantitative advantages of RNA-seq. Of the work that is presented I have almost no concerns and several interesting findings, especially the CHF and other immune-related findings. My concerns and request for major revision are focused on what is not presented. In particular:

1) The source of the S. mansoni parasites used is not presented.

2) There is no discussion of potential parasite gene expression differences in a mouse vs human host.

3) The references to previous RNA-seq analysis of Schistosoma gene expression including S. mansoni is quite lacking, only much older literature is cited. In fact, this manuscript fails to cite a previous Berriman bioRxiv analysis of existing RNA-seq data doi: https://doi.org/10.1101/308189

4) Despite the fact that the S. mansoni genome is now on version 8, with version 7 publicly released by the Wellcome Trust for more than a year in WormBase Parasite, the much older assembly version 5 was sued. This seems odd. In fact, the MS has to go to this newer version to highlight a gene of interest that is mis-assembled in version 5.

4) Most significantly, this manuscript only focuses on the data they have generated and fail to include gene expression insights from the rest the lifecycle. It is impossible to declare, as in line 321 "Lung-stage specific" when all stages are not considered. Given that the RNA-seq data for other stages exists and findings from these studies and microarrays exist these new data should be analyzed in that context.

5) Minor comment line 379 "appeared to upregulated" what is the uncertainty?

Reviewer #3: (No Response)

PLOS authors have the option to publish the peer review history of their article (what does this mean?). If published, this will include your full peer review and any attached files.

Reviewer #1: No

Reviewer #2: No

Reviewer #3: No

---

## [Editor Report · Decision Letter 1]

27 Feb 2020

Dear Dr. Berriman,

We are pleased to inform you that your manuscript 'Transcriptome of the parasitic flatworm Schistosoma mansoni during intra-mammalian development' has been provisionally accepted for publication in PLOS Neglected Tropical Diseases.

Before your manuscript can be formally accepted you will need to complete some formatting changes, which you will receive in a follow up email. A member of our team will be in touch within two working days with a set of requests.

Best regards,

Matty Knight, Ph.D

Associate Editor

Walderez Dutra

Deputy Editor

---

## [Editor Report · Acceptance letter]

24 Apr 2020

Dear Dr. Berriman,

We are delighted to inform you that your manuscript, "Transcriptome of the parasitic flatworm Schistosoma mansoni during intra-mammalian development," has been formally accepted for publication in PLOS Neglected Tropical Diseases.

Best regards,

Serap Aksoy

Editor-in-Chief

Shaden Kamhawi

Editor-in-Chief
